# Impacts of past abrupt land change on local biodiversity globally

Martin Jung [1,4]*, Pedram Rowhani [2] & Jörn P.W. Scharlemann [1,3]*

Abrupt land change, such as deforestation or agricultural intensification, is a key driver of biodiversity change. Following abrupt land change, local biodiversity often continues to be influenced through biotic lag effects. However, current understanding of how terrestrial biodiversity is impacted by past abrupt land changes is incomplete. Here we show that abrupt land change in the past continues to influence present species assemblages globally. We combine geographically and taxonomically broad data on local biodiversity with quantitative estimates of abrupt land change detected within time series of satellite imagery from 1982 to 2015. Species richness and abundance were 4.2% and 2% lower, respectively, and assemblage composition was altered at sites with an abrupt land change compared to unchanged sites, although impacts differed among taxonomic groups. Biodiversity recovered to levels comparable to unchanged sites after >10 years. Ignoring delayed impacts of abrupt land changes likely results in incomplete assessments of biodiversity change.

[1] School of Life Sciences, University of Sussex, Brighton, UK. [2] Department of Geography, School of Global Studies, University of Sussex, Brighton, UK. [3] UN Environment Programme World Conservation Monitoring Centre, 219 Huntingdon Road, Cambridge, UK. [4] Present address: International Institute for Applied Systems Analysis (IIASA), Schlossplatz 1, Laxenburg, Austria. *email: jung@iiasa.ac.at; j.scharlemann@sussex.ac.uk

Natural and anthropogenic processes change the terrestrial surface of the Earth[1,2], which have been shown to impact biodiversity[3,4] and ecosystem services[5]. Previous studies have found that differences in land-surface conditions reduce local biodiversity globally[3,6]. However, biodiversity studies often ignore the impacts of past land change[7,8]. Such change encompasses natural (e.g. fires, storms) and/or human-driven (e.g. agricultural expansion, urbanisation) changes in land use and land cover[9,10]. Simulations and experiments have demonstrated that land changes of greater magnitude have larger impacts on the number of species and individuals[11–13]. Yet, few studies have quantified the impacts of land change in the past on local biodiversity globally.

Local biodiversity continues to be influenced by past land change through biotic lags. Biotic lags—including ecological processes such as extinction debt[14–16], colonization credit[17,18] and ecological memory effects[19]—negatively affect the number of species and individuals present within local assemblages[4,16,20], and potentially reduce resilience[13,21]. The impacts of land change on species assemblages through biotic lag depend on species' abilities to persist[22] and recover[23–25]. Past land changes have been shown to cause 'legacies' affecting ecosystems and species to the present day[7,26]. Most previous global studies[18,24,25,27] investigating abrupt land changes in the past have used descriptive study-specific categories of "land changes", e.g. wild fire, flooding or cultivation, thus hindering comparisons among studies, and preventing predictions. To assess the impacts of abrupt land change on local biodiversity more generally, comparable quantitative measures of "land change" are needed.

The availability of time series of satellite imagery enables the detection and quantification of land changes globally[2,28]. Land change can be quantified as abrupt shifts in intra- and inter-annual dynamics of remotely-sensed photosynthetic activity measured by vegetation indices[29,30]. Abrupt shifts in magnitude[8,31,32] and/or trend[33] of photosynthetic activity, and the time passed since such shifts[32,34] are three key attributes of land change[8]. Several algorithms have been developed to detect abrupt land change[35] and measure these attributes. However, attributes of remotely-sensed abrupt land change have never before been used to assess biotic lags in local biodiversity.

Here we investigate the impacts of abrupt land change in the past—defined as the single largest shift in magnitude and/or trend of photosynthetic activity[2,33,36]—on local biodiversity globally. We use data on local biodiversity of global geographic and broad taxonomic coverage from the Projecting Responses of Ecological Diversity in Changing Terrestrial Systems (PREDICTS) database[37]. At each site, where local biodiversity was sampled at one point in time, we assess time series of high spatial resolution (nominal ~30 m) Landsat satellite imagery from 1982 to 2015 for the presence of an abrupt land change (Fig. 1a) and, where detected, we quantify key attributes, i.e., shifts in magnitude, trend and time passed. Using hierarchical analyses, we compare four measures of local biodiversity (species richness, total abundance, evenness and species turnover) between paired sites (5563 sites with and 10,102 without an abrupt land change) from 377 studies (Fig. 1b). We expect that abrupt land changes with larger shifts in magnitude and trend have greater impacts on local biodiversity through biotic lag effects and that with more time passed local biodiversity can recover from the impacts of abrupt land change. We find that local species richness and abundance are reduced by 4.2 and 2%, respectively, and assemblage composition altered at sites with an abrupt land change compared to unchanged sites, although impacts differed among taxonomic groups. Local biodiversity recovered to levels comparable to unchanged sites after more than 10 years.

## Results

**Local biodiversity at sites with past abrupt land changes.** Sites at which an abrupt land change was observed contained on average 4.2% fewer species (Standard Error (SE): 1.3%, $\chi^2 = 10.3$, df = 3, $p < 0.01$), 2% fewer individuals (SE: 1.3%; $\chi^2 = 72.9$, df = 3, $p < 0.001$), and species assemblages were 1% less even (SE: 0.6%; $\chi^2 = 42.8$, df = 3, $p < 0.001$) compared to unchanged sites (Fig. 2). Sites with larger abrupt shifts in magnitude and trend had fewer species and individuals than unchanged sites regardless of direction of abrupt land change (Fig. 2a, c). Sites with > 50% loss or gain in EVI had on average 15.54% (SE: 5.4%) or 10.53% (SE: 3.4%) fewer species, and 10.7% (SE: 3.8%) or 5% (SE: 3%) fewer individuals than unchanged sites (Fig. 2a, c). Compared to unchanged sites, species assemblages were less even at sites with larger abrupt losses in EVI, but not at sites with larger gains in EVI (Fig. 2e). We found similar impacts of shifts in magnitude and trend on species richness ($\Delta$AIC between mixed effect models for magnitude and trend = 3.22, Pearson's $r$ between impacts = 0.71), abundance ($\Delta$AIC = 2.64, $r$ = 0.61), and evenness ($\Delta$AIC = 5.66, $r$ = 0.98).

**Local biodiversity can recover after abrupt land changes.** We hypothesize that with more time passed local biodiversity recovers to levels comparable to unchanged sites. In line with our expectation we found that sites with an abrupt land change up to 5 years before biodiversity sampling had on average 6.6% fewer species (SE: 1.8%), 3% fewer individuals (SE: 1.8%) and were 2% less even (SE: 0.1%) than unchanged sites (Fig. 2b, d, f). After more than 10 years had passed, biodiversity measures were comparable to unchanged sites (Fig. 2b, d, f), except for local species richness at sites with positive shifts in magnitude or trend (−4%; Fig. 2b). Overall, we found similar impacts of shifts in magnitude and trend and varying time passed for species richness ($\Delta$AIC between mixed effect models for magnitude and trend = 2.85, Pearson's $r$ between impacts $r$ = 0.66), abundance ($\Delta$AIC = 2.46, $r$ = 0.42), and evenness ($\Delta$AIC = 3.03, $r$ = 0.65).

**Abrupt land changes affect composition of assemblages.** Species assemblages at sites with larger abrupt shifts in magnitude were less similar in composition to unchanged sites (Fig. 3a, c). Especially sites with a shift in magnitude of >50% EVI loss or gain were on average less similar (−0.12 and −0.03 proportion of shared species for loss and gain in EVI, respectively) in assemblage composition to unchanged sites (Fig. 3a). Furthermore, the composition of species assemblages was most dissimilar to unchanged sites if an abrupt land change occurred less than 5 years before biodiversity sampling (Fig. 3b, d). After more than 5 years had passed between an abrupt land change and biodiversity sampling, species assemblages were on average more similar in composition (0.04 and 0.001 proportion of shared species for loss and gain in EVI, respectively) to unchanged sites (Fig. 3b). The composition of species assemblages was on average more similar among sites of comparable shifts in magnitude or with time passed (diagonals in Fig. 3a, b) relative to unchanged sites. The impacts of abrupt shifts in magnitude were broadly comparable to shifts in trends although negative shifts in trend impacted assemblage composition more (Supplementary Fig. 4).

**Impacts of abrupt land changes vary among taxonomic groups.** Sites with a positive shift in magnitude had significantly fewer species of plant (−9.5%, Wald test: $z = -4.75$, df = 613, $p < 0.0001$), bird (−4.2%, $z = -2.36$, df = 605, $p = 0.018$) and reptile (−10.4%, $z = -2.05$, df = 95, $p = 0.04$) compared to unchanged sites (Fig. 4a). Particularly sites with a negative shift in trend had significantly fewer species of plant (−5.8%, $z = -2.37$, df = 918,

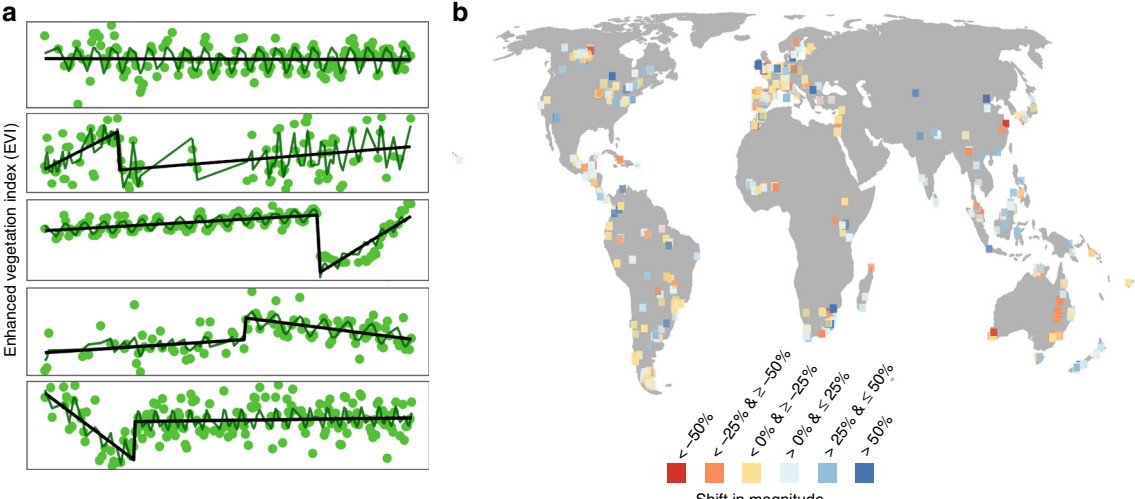

**Fig. 1** Examples of abrupt land change and distribution of sites. **a** Remotely-sensed time series of monthly enhanced vegetation index (EVI; green points) of the same duration at an unchanged site (top panel) and at four sites with an abrupt shift in magnitude (i.e., gain or loss in EVI). Linear (black lines) and seasonal (dark green lines) fits of the change detection algorithm[36] are shown. **b** Location of 5563 sites from 377 studies in the PREDICTS database[37] with an abrupt land change in the monitoring period (since 1982) of the Landsat 4–8 missions with a relative shift in magnitude (location of sites with shifts in trend see Supplementary Fig. 1). For ease of viewing, the location of 10,102 sites without an abrupt land change have been omitted. Map shown in Eckert IV equal-area projection.

$p = 0.002$, Fig. 4a) and fungi ($-29.5\%$, $z = -2.61$, df $= 44$, $p = 0.009$), and fewer individuals of fungi ($-18.37\%$; $z = -2.4$, df $= 43$, $p = 0.016$, Fig. 4b) compared to unchanged sites. The number of individuals and assemblage evenness was overall lower at sites with an abrupt land change compared to unchanged sites, although amphibian and mammal abundance, as well as evenness of flying insects, were higher at sites with an abrupt land change (Fig. 4b, c). For most taxonomic groups, except fungi and reptiles, there was little difference between the impacts of shifts in magnitude and trend on biodiversity measures (Fig. 4).

## Discussion

We found species assemblages to be negatively impacted by past abrupt land change. Larger changes on land caused greater reductions in local biodiversity (Fig. 2a–c) regardless of whether shifts in magnitude or trend of photosynthetic activity (EVI) were positive or negative, suggesting general impacts of past abrupt land change on biodiversity[11,13] likely caused by biotic lag effects[4,17,19]. Abrupt land changes with large (>50%) losses or gains in EVI have caused immediate and time-delayed local extinctions[16,38,39], and reduced the abundance and dominance of persisting species (Fig. 2b, c), which may ultimately affect ecosystem functioning[5,13]. Local biodiversity at sites with an abrupt land change recovered to levels comparable to unchanged sites after >10 years[23,24], although local species richness did not recover at sites where EVI had increased (Fig. 2d). Abrupt land changes can alter the composition of species assemblages with early colonizing and non-native species often outperforming or replacing many persisting species[40–42], which could explain the observed impacts on species assemblage evenness (Fig. 2c) and compositional similarity (Fig. 3). However, to more fully disentangle the impacts of abrupt land change on local biodiversity, before-after, control-impact (BACI) estimates of biodiversity are needed[43], which are currently unavailable globally. Overall, our results suggest that abrupt land changes in the past continue to influence present species assemblages globally.

What drives abrupt land change events? Abrupt land change, identified by shifts in magnitude and/or trend of photosynthetic activity, can be caused by anthropogenic deforestation[31], land

intensification[44,45], or degradation[46,47], or by natural factors such as climatic anomalies[48], nutrient deposition, or $CO_2$ fertilization[49]. Most PREDICTS sites were modified by humans[37,50] and it is therefore likely that most detected land changes were caused by humans. Future studies should attempt to distinguish and disentangle the impacts of natural and anthropogenic abrupt land changes[51] and investigate whether different drivers of land change, such as agricultural conversion or natural fires, have differential impacts on local biodiversity. Furthermore, the duration of land change, e.g. brief leaf die-off vs permanent removal of vegetation, and how this differentially impacts species may be worthwhile investigating[8].

Detecting and quantifying abrupt land changes is challenging. Here, we focussed on detecting abrupt land change as shifts in magnitude or trend[36], but not all land change is abrupt[52] or— such as understory thinning and selective logging—can be detected in time series of remotely-sensed photosynthetic activity[53,54]. Similar to previous studies we assessed only the impact of the single largest shift in magnitude or trend[2,33], while different sequences of land change may also affect local biodiversity[8]. Future studies quantifying abrupt land change globally could benefit from better access to, or fusion of, available satellite data to attain higher temporal and spectral resolution[55,56].

In conclusion, we demonstrate that compared to unchanged sites local biodiversity is considerably reduced because of abrupt land changes in the past, potentially affecting the stability and functioning of ecosystems[13]. Ignoring delayed biodiversity responses to abrupt land changes means that contemporary biodiversity changes, loss and recovery, are underestimated[14,57]. Conservation practitioners need to consider the impacts of biotic lag effects to ensure global and regional assessments (e.g. those by the Intergovernmental Science-Policy Platform on Biodiversity and Ecosystem Services [IPBES]) fully capture biodiversity change[57]. Remote sensing can assist in quantifying attributes of abrupt land change over large spatial and temporal scales. Our analytical framework can be expanded to assess spatial prioritization of habitat restoration plans or to support scenario-based modelling[23] to predict the impacts of abrupt land change on local biodiversity.

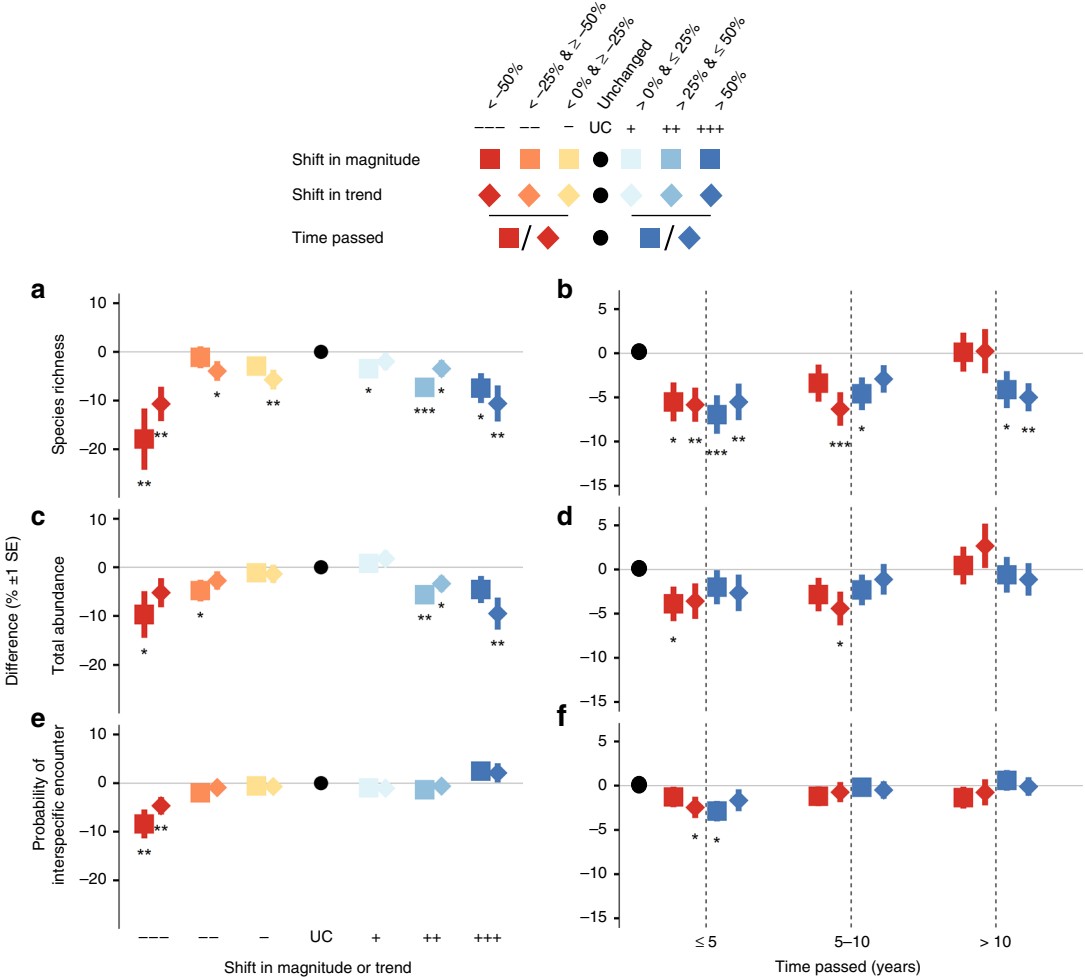

**Fig. 2** Local biodiversity impacts varied with attributes of abrupt land change. Differences in **a, b** local species richness, **c, d** total abundance, and **e, f** the probability of interspecific encounter (PIE) at sites with an abrupt land change (squares, diamonds) relative to unchanged sites (black circles). **a, c, e** Estimates are given separately for shifts in magnitude (squares) and in EVI trend (diamonds). **b, d, f** Impacts on biodiversity measures of time passed between an abrupt land change and sampling of biodiversity for EVI shift in magnitude (squares) or for trend differences (diamonds). Separate models were fitted for shifts in magnitude and in trend relative to unchanged sites (points). Error bars show fitted standard errors and asterisks statistical significance (*$p < 0.05$, **$p < 0.01$, ***$< 0.001$) from the hierarchical mixed effects models. For number of sites and studies for each bin and biodiversity measure see Supplementary Figs. 2, 3 and Table 1. Source data are provided as a Source Data file.

## Methods

**Biodiversity data**. We used published data from the Projecting Responses of Ecological Diversity In Changing Terrestrial Systems (PREDICTS) database[37], which includes species' presence and abundance data from 'studies' with at least two spatially-explicit 'sites', information on the date of sampling, and local land-use and/or land-use intensity[37]. We simplified the original PREDICTS land use and land-use intensity information[37,58] by allocating each site to one of three broad land-use categories: primary vegetation (PV, i.e. primary [non-] forest), secondary vegetation (SV, i.e. mature, intermediate, young and indeterminate age secondary vegetation) or human-dominated vegetation (HDV, i.e., plantation forest, cropland, pasture, urban). Studies were grouped into eight broad taxonomic groups based on the sampled taxa: plants, fungi, ground-dwelling invertebrates (e.g., soil-fauna, snails, beetles), flying invertebrates (e.g., butterflies, bees, dragonflies), amphibians, reptiles, birds or mammals.

We assessed four measures of local biodiversity that complement each other and have previously been shown to be sensitive to abrupt land change[12,27]. For each site in the PREDICTS database, we calculated within-sample species richness and, where data on abundance were available, $\log_{10}$ total abundance of individuals, adjusted by sampling effort following Newbold et al.[59]. After visual inspection, we removed one outlier study (a study of understory plants, ID "DL1_2012__CalvinoCancela") from further analyses because of its unique way of measuring abundance (using biovolume). As a measure of assemblage evenness, we calculated the arcsine square root transformed probability of an interspecific encounter (PIE), which quantifies the probability of two individuals randomly chosen from an assemblage representing different species[60]. As a measure of turnover in species assemblage composition within studies, we calculated the Sørensen similarity index among spatial pairs of sites within each study and land-

use category[61]. All biodiversity measures were calculated using R code available on GitHub (https://github.com/Martin-Jung/PastDisturbance).

Species assemblages were sampled at various spatial extents defined by each study's sampling method and land use. Following the PREDICTS data curation protocol, we assumed the allocated land use to be dominant within the reported sampling extent (maximum linear extent [MLE], in meters) of each site[37,58]. For studies without reported MLE (4779 sites, 18.3% of all sites), we used either the mean MLE for each taxonomic group and corresponding sampling method, e.g., mist netting, pitfall trapping, or the mean MLE within the same taxonomic group. To test whether these interpolated MLEs are consistent among taxonomic groups and sampling methods, we randomly removed 25% of the reported MLEs and found the interpolated MLEs to be reasonably correlated (Pearson's $r = 0.73$, $p < 0.001$). We included all studies with a MLE $\leq 3000$ m (98.3% of all sites), approximately 100 times the nominal resolution (~30 m) of the remotely-sensed data used in this study, and removed four studies with sites located in water (rivers, coastal areas or ponds), identified by intersecting all sites with a global permanent water surface mask[62], as a precaution as sites within these studies likely have low positional accuracy. We cannot rule out that some PREDICTS sites have imprecise coordinates, although we have no evidence or reason to suspect any systematic bias, taxonomically or geographically, in the coordinate accuracy or precision that could substantially affect our findings. For 96.7% of all sites in the PREDICTS database, coordinates were obtained from publications or supplied by the authors of the original studies while for the remaining studies, coordinates were worked out using the information provided in publications, followed by a detailed assessment of coordinate accuracy for all sites as part of the PREDICTS database curation[58]. To spatially link species assemblage with remote sensing data, we calculated a square buffer with the study-specific MLE as side lengths (MLE$_{median} = 70$ m; $Q_1 = 30$ m,

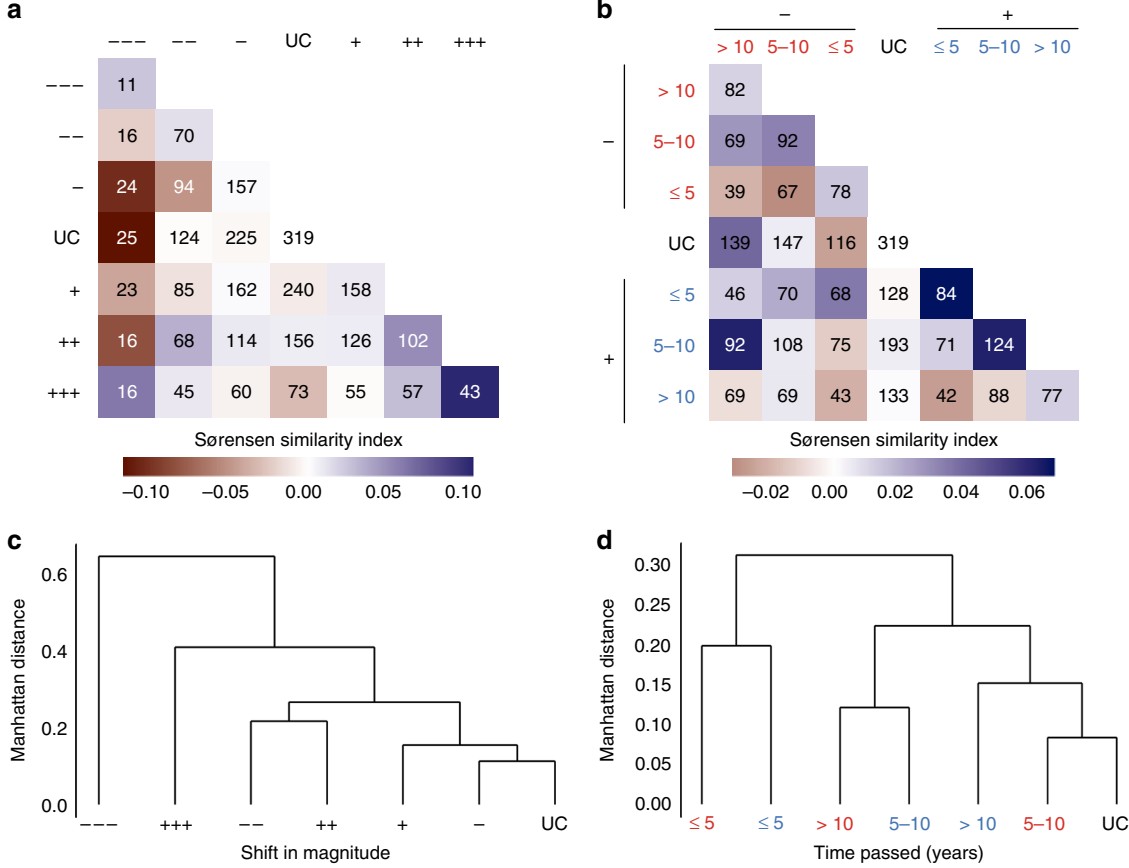

**Fig. 3** Reduced compositional similarity between sites with and without abrupt land change. Mean similarity in species assemblage composition (Sørensen similarity index) calculated between pairs of sites within the same study and land-use category without (UC) and with an abrupt land change of (**a, c**) varying shifts in magnitude, or (**b, d**) loss or gain in EVI (minus signs/red and plus signs/blue) and time passed between abrupt land change and biodiversity sampling (axis labels as in Fig. 2); Colours, from purple to brown (in **a, b**), indicate more or less similar assemblage composition with unchanged sites on average relative to comparisons among unchanged sites. Numbers (in **a, b**) indicate the total number of studies for which pairwise comparisons between sites could be made. All estimates are transformed relative to the compositional similarity between pairs of sites without a land change (UC − UC). (**c, d**) Dendrograms show hierarchical clustering of all pairwise similarities based on the average Manhattan distance between pairs of sites; sites with more similar assemblage composition are in branches of closer proximity.

$Q_4 = 200$ m) around each site's coordinates as the smallest area that fully captures all grid cells of varying sampling units (e.g., point counts, line transects). Square buffers around site coordinates likely capture the area sampled best as the sampling layout of the majority of studies was rectangular.

**Remote sensing data.** We used land-surface reflectance products derived from the sensors of the Landsat 4 (1982–1993), 5 (1984–2012), 7 (1999–ongoing), and 8 (2013–ongoing) missions available within Google Earth Engine (GEE)[63], based on raw United States Geological Service Landsat Collection images (Tier 1) to calculate the Enhanced Vegetation Index (EVI, as two-band version[64]) as a proxy of photosynthetic activity. We masked all cloud-covered grid cells (~30 m nominal resolution) using the cloud-detection output in the 'cfMask' band[65] and removed occasional snow- and water-covered grid cells, i.e. those with negative EVI values. All data preparation and extraction were performed within GEE[63].

For each Landsat image and PREDICTS site we calculated the mean EVI within the rectangular buffer ($\bar{y}$) and extracted time series of all EVI values. We removed outliers introduced by satellite sensor errors, missed cloud shadows or bad quality estimates by calculating the absolute difference of all $\bar{y}$ values from the median absolute deviation (MAD) per EVI time series[66]. EVI values more than a conservative threshold of two units of deviation away from the MAD or in the top 1% of all MAD estimates were set to NA[66]. Time series of EVI data were temporally aggregated to monthly maximum value composites to ensure equal intervals between data points and to reduce the amount of noise and missing data. Because of the ongoing consolidation of the global Landsat archive[55], there can be periods of consecutively missing data, particularly before the launch of Landsat 7 in 1999 (Supplementary Fig. 5a). We truncated time series with gaps of ≥5 years of consecutively missing data, which might affect the precision of land change attribute calculations, by truncating these time series to include only the years from 1999 onwards in subsequent analyses (see Supplementary Fig. 5b). In total 25,656 sites had suitable EVI time series, with an average 18.83 (±6.7 SD) years

duration containing on average 1.82 years (±1.57 SD) of consecutively missing data.

**Abrupt land change detection.** To identify the presence of abrupt land change and its attributes in EVI time series, we used the Breaks For Additive Season and Trend (BFAST) algorithm[34] modified to work with missing data and optimized to find the single most influential abrupt land change in a time series[33]. BFAST accurately detects abrupt land changes[31,36] by using a multiple regression model to estimate both trend and seasonal components of a time series:[33] $\bar{y}_t = \alpha_s + \beta_s t + \sum_{p=1}^{k} \gamma_p \sin\left(\frac{2\pi pt}{h} + \delta_p\right) + \varepsilon_t$, where $\bar{y}_t$ is the mean EVI at time $t$, $s$ the segment in the time series, $\alpha$ the intercept, $\beta$ the slope (i.e., trend), $p$ and $k$ the order of the seasonal term ($k = 2$), $\gamma$ the amplitude, $\delta$ the phase and $\varepsilon$ the residual error. The expected frequency to detect an abrupt land change in a time series is determined by $h$ and, following previous studies[36,67], was set as the ratio of the number of data points per year (12 months) to the total length of the individual time series (in months). Whenever the inclusion of the seasonal component caused the model to fail to converge (17% of all fitted models), we removed the seasonal component by time series decomposition ('stlplus' package[68]) prior to fitting BFAST with a trend component only. BFAST detects abrupt land change when model residuals depart significantly ($p < 0.05$) from a statistical boundary[69]. To test for significant departure we used two complementary approaches[36,67,69] using first, a moving sum of residuals (MOSUM) test within the monitoring period (determined by $h$) and second, an information-theoretic approach, the Bayesian Information criterion (BIC). All BFAST models were fitted using the 'bfast' package (ver. 1.5.7) in R (ver. 3.5)[36,70].

For the single most influential abrupt land change detected in each time series, we calculated the relative shift in magnitude as the immediate change in EVI $\left[\frac{(\hat{y}_j - \hat{y}_{j-1})}{|\hat{y}_{j-1}|}\right]$, where $\hat{y}_j$ is the first monthly estimate of $\bar{y}$ predicted by the BFAST model

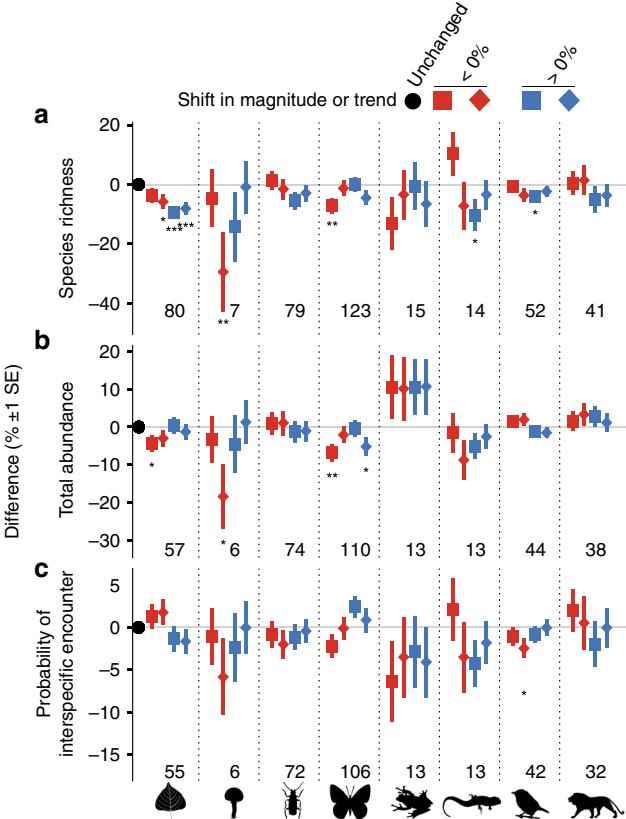

**Fig. 4** Abrupt land change affects taxonomic groups differently. Difference in **a** local species richness, **b** total abundance, and **c** the probability of interspecific encounter for taxonomic groups (plants, fungi, ground-dwelling invertebrates, flying invertebrates, amphibians, reptiles, birds, and mammals) between sites with and without an abrupt land change. Separate models were fitted for taxonomic groups comparing sites with shifts in magnitude (squares) and trend differences (diamonds), where colours indicate negative (red) and positive (blue) direction, to sites without abrupt land change (black circles, grey line). Error bars show standard errors and asterisks indicate statistical significance (*$p < 0.05$, **$p < 0.01$, *** $< 0.001$) from the hierarchical mixed effects models. Numbers give the sample size as the number of studies included per taxonomic group. Source data are provided as a Source Data file. Species silhouettes from http://phylopic.org.

after an abrupt land change has been identified and $\hat{y}_{j-1}$ the predicted estimate one month before], the difference in linear trend as increase/decrease in EVI before and after the abrupt land change ($\beta_{after} - \beta_{before}$, where $\beta_{after}$ and $\beta_{before}$ are the predicted linear trends in EVI from the BFAST model, before and after the abrupt land change), and the time passed (in months, $t_n - t_j$) between the date of the abrupt land change ($t_j$) and the start of biodiversity sampling ($t_n$). Attributes of abrupt land change are grouped into bins as follows (Supplementary Fig. 2 and Table 1): for shifts in magnitude (>50%, >25% and ≤50%, and ≤25% EVI loss or gain, Supplementary Fig. 2a), for shifts in trend (0.01, 0.05, and >0.05 lower or higher EVI trend change, Supplementary Fig. 2b) and time passed (<5, 5-10, and >10 years ago, Supplementary Fig. 2c). The three attributes of abrupt land change were only marginally correlated among each other (mean Pearson's $|r| < 0.07$, Supplementary Fig. 6). Sites without an abrupt land change detected by BFAST are referred to as "unchanged" sites (UC) and all studies containing only unchanged sites (10,196 sites of 262 studies) were excluded from further analyses.

**Statistical analyses**. We built hierarchical models comparing biodiversity measures between paired sites, i.e. those with and without an abrupt land change in the past, from the same study and sampled at the same point in time using the same sampling method to account for differences among studies[71]. Hierarchical generalized linear mixed effects (LME) models were fitted separately for species richness (using a Poisson error distribution), total abundance, and the PIE (using a Gaussian error distribution). For models of species richness we included an observation-level random effect (i.e., site ID) to account for overdispersion[72]. For each LME model we compared several candidate random-effect structures by fitting null models with combinations of different random intercepts and random

slopes to determine the structure with the lowest overall Aikake Information Criterion (AIC). Random effects always included the study ID to account for study-level differences in sampling methods, optionally a spatial block ID in which sites were located, the site's land-use category (PV, SV, HDV), the presence of an abrupt land change (yes|no) as random slope within the study, as well as the studies climatic zone (tropical, arid, temperate or continental climate) according to the Koeppen Geiger classification[73]. Whenever a climatic zone could not be determined (for instance on small islands), we attributed studies to a zone based on latitude and a site's terrestrial biome (1369 sites). The most parsimonious random-effect structure by AIC was identical among response variables and included — besides the study ID — the spatial block and land-use category as random intercept as well as the presence of an abrupt land change as random slope. We included the binned attributes of abrupt land change, e.g. shifts in magnitude, trend, and time passed, as fixed effects in our models with the unchanged sites (UC) as paired reference comparison. Separate models were fitted for each taxonomic group using the direction (positive or negative) of magnitude and trend shifts because of limited data availability. Full LME models were tested for significant differences ($p < 0.05$) from a null model using likelihood ratio tests, while significant differences between bins were approximated by Wald statistics[74]. To compare estimated impacts of a shift in magnitude against shift in trend, we assessed the difference in Akaike's Information criterion (AIC), a difference of ∆AIC < 7 commonly indicating less support of either model being more parsimonious, and furthermore calculated ordinary Pearson correlation coefficients between their effects as models were otherwise not comparable, for instance by conventional maximum likelihood ratio tests, because of equal fixed structures. All statistical tests used were two-sided. All models were fitted using the 'lme4' package (ver. 1.1-14 in R ver. 3.5)[70,74].

To estimate differences in species assemblage composition we calculated the mean compositional similarity (as quantified by the Sørensen similarity index) between all spatial pairs of sites without and with an abrupt land change in the same study and land-use category. To visualize the mean similarity for each land change attribute bin, we performed hierarchical complete-linkage clustering ('hclust' function in R) on Manhattan distances between estimates of compositional similarity transformed relative to the mean difference between pairs of unchanged sites.

**Reporting summary**. Further information on research design is available in the Nature Research Reporting Summary linked to this article.

## Code availability
Codes are available at https://github.com/Martin-Jung/PastDisturbance.

## Data availability
The PREDICTS biodiversity data are publicly available in the Natural History Museum Data Portal (https://doi.org/10.5519/0066354)[37]. All remote sensing data are accessible via Google Earth Engine (https://earthengine.google.com/)[63] and pre-processed time series are deposited on GitHub (https://github.com/Martin-Jung/PastDisturbance). The source data underlying Figs. 2 and 4 and Supplementary Fig 3 are provided as a Source Data file.

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

## Acknowledgements

We thank all PREDICTS data contributors for their biodiversity data, the Land Processes Distributed Active Archive Center (LP DAAC), located at US Geological Survey (USGS)/Earth Resources Observation and Science (EROS) Center for the Landsat data, Google Earth Engine[63] for providing developer access to their cloud-computing facilities, University of Sussex for providing computing facilities. M.J. was funded by a School of Life Sciences, University of Sussex PhD studentship to J.P.W.S. and P.R.

## Author contributions

All authors designed the study. M.J. collected and analysed the data and wrote the first draft of the paper. M.J., P.R. and J.P.W.S. discussed the results and wrote the paper.

## Competing interests

The authors declare no competing interests.
