## [Peer Review File · Nature Communications]

Reviewers' comments:

Reviewer #1 (Remarks to the Author):

This is an exciting and important manuscript that make a major contribution to the biodiversity science. The remote sensing analyses are state of the art, the biodiversity dataset impressive in its global scope, and the major finding regarding the effects of abrupt land changes including their time lags is fascinating. This paper will be highly influential in the field, and spur subsequent research into the exact processes causing both the abrupt land changes and resulting changes in biodiversity.

The manuscript is carefully prepared and well written. I do not see any need for revisions, and recommend acceptance as is.

Reviewer #2 (Remarks to the Author):

The authors identify temporal abrupt changes using remotely sensed temporal series of vegetation indices, and match them with temporal series of biodiversity values in the corresponding locations derived from the PREDICTS database. The authors have certainly put much effort in conducting the analyses, and the overall approach is quite innovative. Results show clear biodiversity responses to vegetation indices changes, which depend on the direction and intensity of the change, and time from the change.

My concern with the article is that is hard to follow and results a bit difficult to interpret. I suggest the authors to put much effort in simplifying as much as possible the text and the communication of the results, especially the figures. I also feel that the discussion does not present particularly novel ideas, and the conclusions is a bit poor. The major problem I see for the interpretation of the patterns is that the authors cannot distinguish between events that affect the photosynthetic activity temporarily (e.g. climatic anomalies, nutrient deposition), from those that change it permanently (e.g. shift to agriculture or any other land cover type), and this greatly limits our ability to intepret the results, especially in relation to the time passed. I guess, one possibility could be to match the results with yearly land cover data, but I'm not sure if land cover data that cover the time frame of the study are available. I'm not expert in remote-sensing but the methods seem to be ok, I only have some concerns about the stats and find the description unclear in some parts.

I can't find an explanation of how paired sites were selected. What constitute a pair? Is it the distance between two sites?

Lines 186-189: Changes in evenness without knowing what happened or without corresponding info on richness, are tough to interpret. The arrival of pioneer species and the decline of local species can

decrease evenness as well as increasing it, it really depends on the original conditions of the community and if species composition changes.

Line 236: Why did you exclude this outlier? Do you consider it as a possible error? Is it an outlier in the comparison with the paired site or compared to other sites in the sample?

Lines 248-250: I don't understand this.

Lines 251-255: Remote sensing data have a resolution of 30m, and you included studies with a MLE<3000 m, and calculated the remote sensed values within a buffer that corresponds to MLE. This means a max of ~3km. What is the precision of the coordinates included in PREDICTS? 3 km is really high resolution, and from my experience most of the cases coordinates from empirical studies are quite uncertain (pointing to a locality, park, etc.), or uncertainty is not reported at all. What is the risk of mismatch between what you measure and the biodiversity values?

Line 256: What is a "rectangular" buffer? and why rectangular and not circular? It seems more prone to errors to me.

Line 330: Why are you using "presence of an abrupt land change" as random and not fixed factor if this is actually what you want to measure?

Line 336: How can you use "presence of an abrupt land change" as random slope, if this is a boolean variable?

Line 337: What do you mean by "binned attributes of abrupt land change"?

Line 344: I have never seen a $\Delta AIC < 7$ considered as "little improvement in model fit". Normally, according to Burnham and Anderson, a ΔAIC between 4 and 7 correspond to "considerably less support".

Further, I don't understand why you calculate Pearson correlation coefficient: "models were otherwise not comparable because of equal fixed structure". Why? In one case you have shift in magnitude and in the other shift in trend, right? I don't understand.

Line 131: "years before"?

Fig. 1. It would be clearer if you matched the panels in (a) with the corresponding combination of --+ symbols in (b). I suggest having 4 panels in (a), one for each of the combinations described in (b).

Fig. 2. is hard to read. Including a visual legend with symbols colors embedded in the figures would facilitate the interpretation.

Fig. 3. I don't understand the figure. The figure is supposed to show the similarity between changed and unchanged sites. However I don't understand why you have the minus and plus symbols on both axes (as they should only refer to the changed situation).

Fig. 4. The Y label of panel C doesn't correspond to the definition of C in the legend. After reading the methods I understood this was actually the same, but the reader should be able to understand it from the figure itself, especially if the format of the article has the methods at the end. Here too, I would include a visual legend in the figure for symbols and colors.

Numbers in the quadrants are confusing, as normally values here correspond to the color, while here represent the sample size.

Reviewer #3 (Remarks to the Author):

General comments

Although the number of studies showing delayed responses of biodiversity (including e.g. land use legacy effects, climatic extinction debts, etc.) to environmental change is increasingly large, studies reporting global impacts of past land change on different facets of biodiversity at local scales are still scarce. This study evidences how abrupt land changes in the past affect current local species richness, abundance and community composition. In particular, the authors used a comprehensive biodiversity dataset created in the context of the PREDICTS project which, according to its webpage, has over 2.5 million biodiversity records from over 21,000 sites covering more than 38,000 species. Nowadays, the increasing availability of cloud-computing platforms (such as GEE) offers an unprecedented opportunity to develop global environmental assessment at local scales. The authors took advantage of this new platforms to analyze land change at local scale globally. In particular, the authors compiled a huge amount of Landsat images to compute vegetation indices (EVI) time series at site scale for a 40-year period (at approx. 30-m resolution). Based on their results, the authors conclude that ignoring delayed impact of abrupt land change will result in incomplete assessment of biodiversity change, which is especially timely and relevant given the global and regional assessments that have been carrying out by the IBPES. I therefore believe that this study will be of the interest of the environmental science community.

From a more methodological viewpoint, the authors calculated several metrics: within-sample species richness, total abundance adjusted by sampling effort, and measures of assemble evenness and turnover. I think that the metrics are appropriate but more information about the R package (and function) used to compute e.g. PIE (Simpson's evenness index) or Sorensen indices is required in order to ensure reproductivity. The authors also applied different filters to deal with common limitations for Landsat time series, which are mostly related to cloud cover and missing data before Landsat 7 was launch (i.e. before 1999). In this regard, it is still not crystal clear to me if the authors finally considered data from 80's or if they truncated time series to include only the years from 1999 onwards in subsequent analyses, and if this truncation was applied to all sites or only to those affected by missing data or high cloud cover. The authors also removed outliers, masked clouds and snow or water-covered grids. However, images obtained from Landsat 7 ETM+ sensor from 2003 onward has a well-known problem associated with the satellite's scan-line corrector fail. The scan-line corrector is a device on the satellite that keeps the scan lines parallel to each other. Without the scan-line corrector, the scan lines are mis-aligned and there are wedge-shaped data gaps in the image (see <https://landsat.usgs.gov/slc-products-background>). USGS offers a number of different procedures for filling-in the data gaps, and also images provided by Landsat 5 for that period (still operative at that time). This is an important limitation, large areas within the Landsat scenes acquired by the ETM plus sensor from 2003 onward might be strongly affected by the scan-line corrector fail, affecting the interannual EVI trends. However, the authors do not mention this problem in the text nor provide details about how they overcame this important limitation.

To identify abrupt land changes and its attributes in EVI time series, the authors used the Breaks For Additive Season and Trend (BFAST) algorithm. Although I am not personally very familiar with this method, BFAST is widely used by the remote sensing community for detecting and characterizing change within time series. I agree with the bins defined by the authors to group the different attributes of the abrupt land change, as they are understandable, informative and not correlated among each other. Figures are easy to read.

The authors applied hierarchical generalized liner mixed models. I think that the method is appropriate given that biodiversity data from several sites within the same climatic or landscape "context" may be correlated. Did the authors test that possible correlation among error components? Or it was simply assumed, given the nature of the data? Please clarify. In my opinion, it would be worth briefly justifying the suitability and advantages of this approach in the text.

Regarding data availability, biodiversity data are publicly available, Landsat images accessible via GEE, and pre-processed time series will be deposited on GitHub after publication. The author will also make the code available after publication.

Minor comments

Line 35. What the authors mean by "present difference ... reduce local biodiversity".

Line 40-43. The authors mention several processes related to biotic lags caused by past land change. However, they don't mention the concept "land-use legacy", which was in fact defined as the abiotic and biotic properties found at a site due to the influence of land-use history. I think it is worth mentioning.

Lines 129 and 133. These are the values returned by Sorensen index? please clarify.

Lines 219. The authors should provide a briefly description of the current dataset here, and more details in Supplementary material. For instance, how many studies, sites and species per taxon includes the dataset used in this study? According to the website, the project is still open to contributions, so the dataset will evolve as new data will be incorporated.

Lines 278-282. It is not clear to me if the authors only worked with time-series data from 1999 onwards for all sites or only truncated the time series for those sites with gaps of 5 years of missing data.

I hope my comments help improve the manuscript,

All the best,

Adrián Regos.

Wednesday, 15 May 2019

Recommendation by the Associate Editor (Izzadora Andrew):

Your manuscript entitled "Impacts of past abrupt land change on local biodiversity globally" has now been seen by 3 referees. You will see from their comments below that while they find your work of interest, some important points are raised. We are interested in the possibility of publishing your study in Nature Communications, but would like to consider your response to these concerns in the form of a revised manuscript before we make a final decision on publication.

We thank the associate editor for sending our manuscript to reviewers and the three reviewers for their time and constructive comments. Below we respond (text in bold) to each reviewers' comments (text in non-bold). Please also see the tracked changes in our revised manuscript.

Reviewer(s)' Comments to Author:

Reviewer: #1

This is an exciting and important manuscript that make a major contribution to the biodiversity science. The remote sensing analyses are state of the art, the biodiversity dataset impressive in its global scope, and the major finding regarding the effects of abrupt land changes including their time lags is fascinating. This paper will be highly influential in the field, and spur subsequent research into the exact processes causing both the abrupt land changes and resulting changes in biodiversity.

The manuscript is carefully prepared and well written. I do not see any need for revisions, and recommend acceptance as is.

- 1. We thank Reviewer #1 for taking time to review our manuscript, their kind words and favourable review.**

Reviewer: #2

The authors identify temporal abrupt changes using remotely sensed temporal series of vegetation indices, and match them with temporal series of biodiversity values in the corresponding locations derived from the PREDICTS database. The authors have certainly put much effort in conducting the analyses, and the overall approach is quite innovative. Results show clear biodiversity responses to vegetation indices changes, which depend on the direction and intensity of the change, and time from the change.

My concern with the article is that is hard to follow and results a bit difficult to interpret. I suggest the authors to put much effort in simplifying as much as possible the text and the communication of the results, especially the figures. I also feel that the discussion does not present particularly novel ideas, and the conclusions is a bit poor.

- 2. We thank reviewer #2 for taking time to review our manuscript and providing helpful suggestions. We like to clarify that our analyses presented did not analyse “temporal series of biodiversity values”. Instead, we used time series of a remotely-sensed vegetation index and related these to local biodiversity measures sampled at one point in time from the PREDICTS database. We have added “at one point in time” when we introduce the PREDICTS data (L62-63) “At each site, where local biodiversity was sampled at one point in time, we assessed time series of high spatial resolution (nominal ~30m) Landsat satellite imagery from 1982-2015...” and reworded the sentence in the Methods to read “We built hierarchical models comparing biodiversity measures between paired sites, i.e. those with and without an abrupt land change in the past, from**

the same study and sampled at the same point in time using the same sampling method to account for differences among studies” to the manuscript to emphasize this (L 332-335). Based on this reviewer’s comments and reviewer #1 comments stating that “the manuscript is carefully prepared and well written”, we have rewritten parts of the manuscript, especially the Results and Conclusion sections, and revised the figures, so as to simplify and help understanding (see responses to specific comments below).

The major problem I see for the interpretation of the patterns is that the authors cannot distinguish between events that affect the photosynthetic activity temporarily (e.g. climatic anomalies, nutrient deposition), from those that change it permanently (e.g. shift to agriculture or any other land cover type), and this greatly limits our ability to interpret the results, especially in relation to the time passed. I guess, one possibility could be to match the results with yearly land cover data, but I'm not sure if land cover data that cover the time frame of the study are available. I'm not expert in remote-sensing but the methods seem to be ok, I only have some concerns about the stats and find the description unclear in some parts.

- 3. In our manuscript we followed the definition of land change by Turner et al. (2007), without specifically differentiating between “persistent” or “short-lived” land-use and/or land-cover changes (Lambin and Geist 2006, Turner et al. 2007). We agree that it would be interesting to differentiate between impacts of temporary and permanent land changes on local biodiversity (following the terminology by Lambin and Geist [2006]), however such analyses are beyond the scope of this manuscript. Not disentangling the causes of land change does not limit the interpretation of the results.**

I can't find an explanation of how paired sites were selected. What constitute a pair? Is it the distance between two sites?

- 4. “Paired sites” in the PREDICTS modelling framework refers to sites within the same study sampled at the same time using the same methods (Purvis et al. 2018) that have some form of factorial contrast such as differing land-use/land-use intensity (Newbold et al. 2015), are within or outside protected areas (Gray et al. 2016), or with or without a past abrupt land change (this study). This is mentioned in line 62-63 and we have furthermore added additional explanation to the methods (L 332 -335).**

Lines 186-189: Changes in evenness without knowing what happened or without corresponding info on richness, are tough to interpret. The arrival of pioneer species and the decline of local species can decrease evenness as well as increasing it, it really depends on the original conditions of the community and if species composition changes.

- 5. In this manuscript we assessed the impacts of past abrupt land change on multiple biodiversity measures. Using data from paired sites, we found that species richness, abundance and evenness were on average lower (Figure 2) and that sites with abrupt land changes in the past have an altered species assemblage composition less similar (Figure 3) compared to unchanged sites. We used multiple biodiversity measures, i.e. richness, abundance, evenness and turnover, to highlight the differential impacts of land change on different aspects of biodiversity recognizing that a single measure alone could be misleading (Smith and Wilson 1996, Su et al. 2004, Hillebrand et al. 2018). With all biodiversity measures in this manuscript, we only compared relative and not absolute differences between sites. We agree with the reviewer that evenness is less straight forward to interpret than species richness or abundance, especially when analysing spatial differences in biodiversity without knowing the “original” state of biodiversity before an abrupt land change occurred (also see De Palma et al. [2018]). To fully assess the impacts of land change on biodiversity, before-after control-impact studies would be needed (De Palma et al. 2018), however such data are currently unavailable globally. We have added this as a limitation to the discussion (L 195-197).**

Line 236: Why did you exclude this outlier? Do you consider it as a possible error? Is it an outlier in the comparison with the paired site or compared to other sites in the sample?

6. **The PREDICTS project collates local biodiversity data across taxonomic groups and sampling methods. The excluded study is an outlier in the way this study has measured abundance (i.e. the biovolume of understory plants) compared to all other studies in the PREDICTS database and has been commonly removed in other analyses on the same data (A. Purvis, personal communication). We do not consider this study an error and only removed the study as a precaution. It should be noted that including this study in the analysis does not affect our main results or conclusions.**

Lines 248-250: I don't understand this.

7. **Not all studies and sites (18.3% of all sites) within the PREDICTS database have reported estimates of the areal extent of sampling, as measured by maximum linear extent (MLE). Following a previous study (Jung et al. 2018) we filled the missing MLE estimates with the mean MLE observed in studies of the same taxonomic group and sampling method (L 252-255). This assumes that sampling extents are comparable within taxonomic groups. For example, the area sampled using mist netting for birds is likely similar among bird mist netting studies, likewise the area sampled using pitfall traps for invertebrates is similar among pitfall trapping studies. We tested the assumption that sampling extent correlates with taxon sampled and sampling method used by removing 25% of reported MLE estimates at random and using the remaining data to fill those randomly removed estimates. We found this interpolation to be reasonably correlated with observed MLE estimates (L 258-259).**

Lines 251-255: Remote sensing data have a resolution of 30m, and you included studies with a MLE<3000 m, and calculated the remote sensed values within a buffer that corresponds to MLE. This means a max of ~3km. What is the precision of the coordinates included in PREDICTS? 3 km is really high resolution, and from my experience most of the cases coordinates from empirical studies are quite uncertain (pointing to a locality, park, etc.), or uncertainty is not reported at all. What is the risk of mismatch between what you measure and the biodiversity values?

We thank the reviewer for this comment that raises an important issue for most broad-scale analyses. In our study, the maximum linear extent (MLE) was chosen specifically to capture the area of the local sampling extent and the land-surface conditions observed before and at the time of biodiversity sampling. The PREDICTS database does not hold information on coordinate precision. 95.6% of the sites included in our analyses from the PREDICTS database have coordinates supplied by the authors or mentioned in the published study. We assumed study authors are reporting site coordinates as precisely and accurately as possible. A MLE of 3000m for the sampling extent is at the extreme right tail of the distribution ($MLE_{\text{mean}} = 412.1 \text{ m} \pm 1661.82 \text{ m SD}$) with only 19 (of 377) studies having MLEs of between 2000 and 3000m. Every effort was made to ensure the coordinates are correct during the data curation of the PREDICTS database (see Hudson et al. 2014, 2017) and also during the data preparation for this study. We checked for, and excluded some, outlier studies, e.g. whose coordinates located them in coastal waters or ponds (L 261-264) and visually inspected (not stated in manuscript) a number of sites to confirm that the reported land-use at the site (e.g. forest or cropland) can be seen in Landsat imagery at the time of sampling.

However we cannot rule out that coordinates in the PREDICTS database are imprecise and we added text in the methods mentioning this as a potential limitation (L 264-265). Coordinate precision is an issue for this and all other analyses that worked with study-

derived coordinates (Newbold et al. 2014, Gray et al. 2016, Jung et al. 2017) at medium to high resolution scales (\sim below 0.5° degree). We argue that (i) averaging across remotely-sensed data within a rectangular buffer (defined by MLE) will compensate for some potential inaccuracies of the coordinates and the mobility of animal species when assessing the impacts of land change and (ii) even if the coordinates for some sites are imprecise, this will contribute to the overall uncertainty in the estimated coefficients and impacts of land change on biodiversity (Figure 2). We have no reason to suspect that there is any bias in coordinate uncertainty for particular land uses/land covers, taxa or geographical regions.

Line 256: What is a “rectangular” buffer? and why rectangular and not circular? It seems more prone to errors to me.

8. In this study we used the PREDICTS maximum linear extent (MLE) instead of the centre coordinates for each PREDICTS site. A MLE is determined by the sampling methods of each study, which can be for example circular (point count) or linear (transect) in shape (see supporting Information of Hudson et al. 2014). Rectangular, or square, buffers can capture image grid cells in close proximity which are not touching a circular buffer (e.g., corners in Figure 1 below). However in most situations the number of grid cells contributing to the mean EVI extracted from the Landsat imagery is identical for rectangular and circular buffers.

Figure 1: Schematic of the difference between centre coordinates for a sampling site (white), a circular buffer (blue) and a rectangular buffer (green) that captures the MLE (radius of the blue circle). Fine dotted lines indicate the underlying grid cells.

Line 330: Why are you using “presence of an abrupt land change” as random and not fixed factor if this is actually what you want to measure?

9. We included the “presence of an abrupt land change” as fixed and random slope (see response 10). We have altered the respective sentence (L 343).

Line 336: How can you use “presence of an abrupt land change” as random slope, if this is a boolean variable?

10. By including the presence of an abrupt land change as random slope per study we ensured that impacts of abrupt land change in the past were allowed to vary at the study level (see L 340-341). Not including this categorical factor as random slope would

constrain studies to share a common slope, potentially inflating Type I and Type II errors (Harrison et al. 2018) as well as loosening variance at the study level. It is generally recommended to allow slopes to vary among groups in mixed effects models if the effect of interest might be different per group (Grueber et al. 2011). Including this categorical factor as a random slope furthermore resulted in a model with the highest support (assessed by AIC, see L 347-349) compared to the next best model that does not allow the impacts of abrupt land change to vary ($\Delta AIC = -2495.49$).

Line 337: What do you mean by “binned attributes of abrupt land change”?

- 11. In preparation for the analyses, we grouped attributes of abrupt land change into categorical bins (see L 320ff.). Those binned attributes of abrupt land change were included as fixed effects in the hierarchical linear model with the unchanged sites as reference comparison (L 351).**

Line 344: I have never seen a $\Delta AIC < 7$ considered as “little improvement in model fit”. Normally, according to Burnham and Anderson, a ΔAIC between 4 and 7 correspond to “considerably less support”.

- 12. We have reworded this respective part, now stating “..., a difference of $\Delta AIC < 7$ commonly indicating less support of either model being more parsimonious” (L 357-358).**

Further, I don't understand why you calculate Pearson correlation coefficient: “models were otherwise not comparable because of equal fixed structure”. Why? In one case you have shift in magnitude and in the other shift in trend, right? I don't understand.

- 13. The AIC only tests whether one model (i.e. shift or magnitude) is more parsimonious and has more support than the other, but AIC does not assess whether both models result in similar estimates (i.e. predicted local loss in species richness). We calculated Pearson correlation coefficients between the impacts of shift in magnitude or trend to test whether predicted impacts are broadly comparable. Conventional maximum likelihood tests are not appropriate for our analyses because both shift in magnitude and trend have equal fixed effects structure (i.e. the same sites having abrupt land changes in the past) thus resulting in a lack of degrees of freedom ($df = 0$). We have added additional explanation to the methods (L 360-361).**

Line 131: “years before”?

- 14. We have reworded this sentence, now reading “... five years before biodiversity sampling” (L 133).**

Fig. 1. It would be clearer if you matched the panels in (a) with the corresponding combination of --+ symbols in (b). I suggest having 4 panels in (a), one for each of the combinations described in (b).

- 15. We thank the reviewer for this suggestion to improve Figure 1. We have modified Figure 1, to provide 5 panels of example EVI time series, showing an unchanged site and four sites with combinations of abrupt gain/loss of EVI and shift in EVI trend.**

Fig. 2. is hard to read. Including a visual legend with symbols colours embedded in the figures would facilitate the interpretation.

- 16. We have increased the size of the data points and text to make the figure more legible. We have not added a visual legend to avoid making the figure appear cluttered. All necessary information about the meaning of the colours and shapes in the figure are fully described in the figure legend. Please note response 23 to reviewer #3 below.**

Fig. 3. I don't understand the figure. The figure is supposed to show the similarity between changed and unchanged sites. However I don't understand why you have the minus and plus symbols on both axes (as they should only refer to the changed situation).

17. We have reworded the figure legend to provide additional explanation. All colours in a) and b) indicate the mean similarity between sites with and without abrupt land change, but were transformed relative to pairs of sites without an abrupt land change, thus providing a baseline comparison (see methods L 363-368 or Figure 2 in Newbold et al. (2015), who used the same approach).

Fig. 4. The Y label of panel C doesn't correspond to the definition of C in the legend. After reading the methods I understood this was actually the same, but the reader should be able to understand it from the figure itself, especially if the format of the article has the methods at the end. Here too, I would include a visual legend in the figure for symbols and colors. Numbers in the quadrants are confusing, as normally values here correspond to the color, while here represent the sample size.

18. Thank you for spotting this mistake in the Y-axis label. We have changed the legend of Fig. 4. To avoid clutter in the figure itself, we have not added a visual legend (cf. response 16) as the meaning of symbols and colours is described fully in the figure legend.

Reviewer: #3

General comments

Although the number of studies showing delayed responses of biodiversity (including e.g. land use legacy effects, climatic extinction debts, etc.) to environmental change is increasingly large, studies reporting global impacts of past land change on different facets of biodiversity at local scales are still scarce. This study evidences how abrupt land changes in the past affect current local species richness, abundance and community composition. In particular, the authors used a comprehensive biodiversity dataset created in the context of the PREDICTS project which, according to its webpage, has over 2.5 million biodiversity records from over 21,000 sites covering more than 38,000 species. Nowadays, the increasing availability of cloud-computing platforms (such as GEE) offers an unprecedented opportunity to develop global environmental assessment at local scales. The authors took advantage of this new platforms to analyze land change at local scale globally. In particular, the authors compiled a huge amount of Landsat images to compute vegetation indices (EVI) time series at site scale for a 40-year period (at approx. 30-m resolution). Based on their results, the authors conclude that ignoring delayed impact of abrupt land change will result in incomplete assessment of biodiversity change, which is especially timely and relevant given the global and regional assessments that have been carrying out by the IBPES. I therefore believe that this study will be of the interest of the environmental science community.

19. We thank reviewer #3 for the time to read our manuscript and providing helpful suggestions.

From a more methodological viewpoint, the authors calculated several metrics: within-sample species richness, total abundance adjusted by sampling effort, and measures of assemble evenness and turnover. I think that the metrics are appropriate but more information about the R package (and function) used to compute e.g. PIE (Simpson's evenness index) or Sorensen indices is required in order to ensure reproductivity.

20. The computation of the individual biodiversity measures was done using custom written R-code and functions which we will make available on GitHub upon

acceptance/publication. We have added a sentence to the Methods section of the manuscript (L 249-250) and note that this is also stated on L 376-378.

The authors also applied different filters to deal with common limitations for Landsat time series, which are mostly related to cloud cover and missing data before Landsat 7 was launch (i.e. before 1999). In this regard, it is still not crystal clear to me if the authors finally considered data from 80's or if they truncated time series to include only the years from 1999 onwards in subsequent analyses, and if this truncation was applied to all sites or only to those affected by missing data or high cloud cover.

- 21. In this study, we aimed to obtain the longest possible characterization of land-surface conditions for every PREDICTS site globally. All data available since 1982 was extracted. However, for sites with time series with long gaps (≥ 5 years) before the start of Landsat 7 (pre 1999), we truncated those time series to avoid such gaps affecting the land change attribute calculations. The final dataset contains both time series from 1982 (where available) and truncated time series with long gaps removed. We have visualized the temporal distribution of all time series in Supplementary Figure 4.**

The authors also removed outliers, masked clouds and snow or water-covered grids. However, images obtained from Landsat 7 ETM+ sensor from 2003 onward has a well-known problem associated with the satellite's scan-line corrector fail. The scan-line corrector is a device on the satellite that keeps the scan lines parallel to each other. Without the scan-line corrector, the scan lines are mis-aligned and there are wedge-shaped data gaps in the image (see <https://landsat.usgs.gov/slc-products-background>). USGS offers a number of different procedures for filling-in the data gaps, and also images provided by Landsat 5 for that period (still operative at that time). This is an important limitation, large areas within the Landsat scenes acquired by the ETM plus sensor from 2003 onward might be strongly affected by the scan-line corrector fail, affecting the interannual EVI trends. However, the authors do not mention this problem in the text nor provide details about how they overcame this important limitation.

- 22. We thank the reviewer for highlighting this issue. The failure of the SLC can create data gaps in Landsat 7 imagery, which we set to NA within Google Earth Engine (GEE). We did not apply any pre-extraction gap filling approaches for the Landsat images, but instead averaged all available EVI values within the rectangular buffer around sites and extracted the average from GEE to use in all further analyses (L 278-279). We conducted a monthly compositing to further reduce the influence of data gaps (L 284-286). This approach closely aligns with the official USGS recommendations for dealing with SLC-off gaps (<https://landsat.usgs.gov/filling-gaps-use-scientific-analysis>). Furthermore, the change detection algorithm employed in our analyses can cope with missing data in time series and we also ensured that data gaps were not extensive (< 5 years) (also see response 21).**

To identify abrupt land changes and its attributes in EVI time series, the authors used the Breaks For Additive Season and Trend (BFAST) algorithm. Although I am not personally very familiar with this method, BFAST is widely used by the remote sensing community for detecting and characterizing change within time series. I agree with the bins defined by the authors to group the different attributes of the abrupt land change, as they are understandable, informative and not correlated among each other. Figures are easy to read.

- 23. We thank the reviewer for this comment. In response to reviewer #2 we have updated the figures to make them even easier to understand (see response 15 & 16).**

The authors applied hierarchical generalized linear mixed models. I think that the method is appropriate given that biodiversity data from several sites within the same climatic or landscape “context” may be correlated. Did the authors test that possible correlation among error components? Or it was simply assumed, given the nature of the data? Please clarify. In my opinion, it would be worth briefly justifying the suitability and advantages of this approach in the text.

24. Most analyses of PREDICTS data use a hierarchical modelling approach that aims to quantify differences between paired sites (also see response 4 above) at the study level (Purvis et al. 2018) and - as the reviewer correctly points out -paired sites from the same study are usually within the same climatic zone and landscape. Previous studies (Newbold et al. 2015, Jung et al. 2018) have shown that there is little correlation among error components within those studies. We have added some text about the hierarchical modelling approach in the Methods (L 331-333). Given the strict word limit we have not added an expansive discussion of the suitability and advantages of the approach to the manuscript, and instead refer readers to Purvis et al. (2018) which provides a more in-depth explanation of the approach.

Regarding data availability, biodiversity data are publicly available, Landsat images accessible via GEE, and pre-processed time series will be deposited on GitHub after publication. The author will also make the code available after publication.

Minor comments

Line 35. What the authors mean by “present difference ... reduce local biodiversity”.

25. We have changed this sentence and removed “present”, now stating “Previous studies have found that differences in land-surface conditions reduce local biodiversity globally “ (line 35).

Line 40-43. The authors mention several processes related to biotic lags caused by past land change. However, they don't mention the concept “land-use legacy”, which was in fact defined as the abiotic and biotic properties found at a site due to the influence of land-use history. I think it is worth mentioning.

26. We thank the reviewer for this comment. While the concept of “land-use legacy” is widely established in the literature, it implies that past “land use” is known. The change detection algorithm in this manuscript is able to detect abrupt land change (see Turner et al. [2007]) as changes in terrestrial photosynthetic activity. While those abrupt land changes can be caused by changes in land use and/or land cover, we did not attempt to differentiate between them (see response 3). We therefore decided to avoid using the terms “land-use” and “land-use legacy” all together in the text.

Lines 129 and 133. These are the values returned by Sorensen index? please clarify.

27. Yes, they measure the difference in the proportion of shared species between sites as quantified by the Sorensen index. We have changed the text to highlight this better (L 130 & L 148-150).

Lines 219. The authors should provide a briefly description of the current dataset here, and more details in Supplementary material. For instance, how many studies, sites and species per taxon includes the dataset used in this study? According to the website, the project is still open to contributions, so the dataset will evolve as new data will be incorporated.

28. In the analyses presented here we used the latest published database from the PREDICTS project (now emphasized in line 225). We refer to the published data paper for detailed information on taxonomic and geographic breadth of the database (Hudson et al. 2017).

Lines 278-282. It is not clear to me if the authors only worked with time-series data from 1999 onwards for all sites or only truncated the time series for those sites with gaps of 5 years of missing data.

29. The latter, i.e. we truncated only those time series with gaps of ≥ 5 years before 1999 (also see response 21 above). In the analysis, all available time series (including those truncated) were used. We have reworded the sentence in the Methods (L 287-290).

References:

- De Palma, A. et al. 2018. Challenges With Inferring How Land-Use Affects Terrestrial Biodiversity: Study Design, Time, Space and Synthesis. - *Adv. Ecol. Res.* 58: 163–199.
- Gray, C. L. et al. 2016. Local biodiversity is higher inside than outside terrestrial protected areas worldwide. - *Nat. Commun.* 7: 12306.
- Grueber, C. E. et al. 2011. Multimodel inference in ecology and evolution: challenges and solutions. - *J. Evol. Biol.* 24: 699–711.
- Harrison, X. A. et al. 2018. A brief introduction to mixed effects modelling and multi-model inference in ecology. - *PeerJ* 6: e4794.
- Hillebrand, H. et al. 2018. Biodiversity change is uncoupled from species richness trends: Consequences for conservation and monitoring (M Cadotte, Ed.). - *J. Appl. Ecol.* 55: 169–184.
- Hudson, L. N. et al. 2014. The PREDICTS database: a global database of how local terrestrial biodiversity responds to human impacts. - *Ecol. Evol.* 4: 4701–4735.
- Hudson, L. N. et al. 2017. The database of the PREDICTS (Projecting Responses of Ecological Diversity In Changing Terrestrial Systems) project. - *Ecol. Evol.* 7: 145–188.
- Jung, M. et al. 2017. Local factors mediate the response of biodiversity to land use on two African mountains. - *Anim. Conserv.* 20: 370–381.
- Jung, M. et al. 2018. Local species assemblages are influenced more by past than current dissimilarities in photosynthetic activity. - *Ecography (Cop.)*. 42: 670–682.
- Lambin, E. and Geist, H. 2006. *Land-Use and Land-Cover Change*. - Springer Berlin Heidelberg.

- Newbold, T. et al. 2014. A global model of the response of tropical and sub-tropical forest biodiversity to anthropogenic pressures. - *Proc. R. Soc. B Biol. Sci.* 281: 20141371–20141371.
- Newbold, T. et al. 2015. Global effects of land use on local terrestrial biodiversity. - *Nature* 520: 45–50.
- Purvis, A. et al. 2018. Modelling and Projecting the Response of Local Terrestrial Biodiversity Worldwide to Land Use and Related Pressures: The PREDICTS Project. - In: *Advances in Ecological Research*. 1st ed.n. Elsevier Ltd., pp. 201–241.
- Smith, B. and Wilson, J. B. 1996. A Consumer's Guide to Evenness Indices. - *Oikos* 76: 70.
- Su, J. C. et al. 2004. Beyond Species Richness: Community Similarity as a Measure of Cross-Taxon Congruence for Coarse-Filter Conservation. - *Conserv. Biol.* 18: 167–173.
- Turner, B. L. et al. 2007. The emergence of land change science for global environmental change and sustainability. - *Proc. Natl. Acad. Sci.* 104: 20666–20671.

Reviewers' comments:

Reviewer #2 (Remarks to the Author):

I'm satisfied with some of the revisions and responses, and I think the text improved in clarity. The manuscript is clearly novel, the methodology is well thought and some results are interesting and thought-provoking. Yet the discussion remains very superficial, and in my opinion this is due to a limited understanding of the causes of change. Further, I'm still doubtful about several aspects. I have the feeling the authors did not take very seriously some of my original comments which I reiterate below.

On the contrary of what stated by the authors "Not disentangling the causes of land change does not limit the interpretation of the results." I think that not knowing what causes the abrupt change in EVI does affect our interpretation, especially in relation to how the community recovers. It makes a huge difference if the change was temporary (e.g. climatic anomaly) or permanent (e.g. land cover change) in the expected effects on the community (richness, abundance and evenness). As I suggested previously, I don't think it would be particularly problematic to assess it using land cover time series.

The authors replied to one of my comment on the uncertainty in coordinate precision:

"We assumed study authors are reporting site coordinates as precisely and accurately as possible."

I have experience with data collection of coordinates from field studies, and I believe this assumption cannot be made. 3000m is nothing when coordinates are expressed as decimal degrees with one decimal only, which is extremely common in field studies. Further, there are obvious rounding errors (rounding minutes in degree-minute-seconds is also quite common), note that field studies report coordinates for different purposes than being used in other analyses. This can be easily checked by plotting the coordinates provided by field studies for small islands or along the coast, which very often fall into the sea. To be on the safe side, I would suggest to repeat the analysis extending the buffer further than the MLE (it can be a sensitivity analysis).

The response "in most situations the number of grid cells contributing to the mean EVI extracted from the Landsat imagery is identical for rectangular and circular buffers" and figure shown in relation to my comment on the rectangular buffer are misleading, as the effect is entirely dependent on the resolution used and the size of the buffer. The larger the buffer and the higher the resolution, the more problematic is the use of a rectangular buffer. That said, the author did not respond to my question. Why a rectangular buffer? A circular buffer is in principle a better approximation.

(Irrespective of what other referees believe) I still find the interpretation of Fig. 2 and 3 unnecessarily complex. It doesn't convey a clear message at glance, it's a lot of information to process (symbols, colours, categories, variables...). To me, a figure that takes too long to be

interpreted and require the reader to check the legend multiple times is not a good way to present the results. In my opinion, the aim of figures is not to provide all possible informations (tables are a better option for that) but to simplify and convey a clear message. Eventually, this is a choice of the authors, but I suggest them to think carefully what to show and how to make it as clear as possible, normally, less is better. Note that you are aiming for a multidisciplinary journal with a very wide audience with different backgrounds.

Remember to change line 251

365: add space

Reviewer #3 (Remarks to the Author):

Thanks to the authors for clarifying and addressing all my comments and concerns. I think that the manuscript can be now accepted for publication as it is.

All the best,

Adrián.

NCOMMS-19-02141A, Impacts of past abrupt land change on local biodiversity globally

Please find our responses to the editor's and reviewers' comments in plain text below.

Reviewers' comments:

Reviewer #2 (Remarks to the Author):

I'm satisfied with some of the revisions and responses, and I think the text improved in clarity. The manuscript is clearly novel, the methodology is well thought and some results are interesting and thought-provoking. Yet the discussion remains very superficial, and in my opinion this is due to a limited understanding of the causes of change. Further, I'm still doubtful about several aspects. I have the feeling the authors did not take very seriously some of my original comments which I reiterate below.

1. We thank the reviewer for reading our revised manuscript and responses to reviewer's comments, and for their positive overall comments on the novelty of the manuscript, the thought-out methodology, and the interest in our results. We address the specific comments in detail below and have revised the manuscript and figures in response.

1) On the contrary of what stated by the authors "Not disentangling the causes of land change does not limit the interpretation of the results." I think that not knowing what causes the abrupt change in EVI does affect our interpretation, especially in relation to how the community recovers. It makes a huge difference if the change was temporary (e.g. climatic anomaly) or permanent (e.g. land cover change) in the expected effects on the community (richness, abundance and evenness). As I suggested previously, I don't think it would be particularly problematic to assess it using land cover time series.

2. The reviewer raises (at least) two separate issues, namely A) what the causes of abrupt land change are and how different causes may impact on biodiversity, and B) whether temporary or permanent land changes affect biodiversity differently.

In response to A): Our study sets out to assess the impacts of abrupt land change on local biodiversity and not what the causes, or the drivers (as they are frequently called), of land change are. Land change can be a combination of changes in land cover and/or land use. The former describes what covers the land (e.g. grassland, forest, water), which can be observed directly using satellites; whereas the latter describes how land is being used by humans (e.g. low intensity grassland, high intensity farmland), which as a process cannot be observed directly using remote sensing. Identifying the drivers of land change is an extremely challenging task, especially at global scales, as the same outcome in terms of changed land may result from several causes. For example, the loss of trees may be caused by human deforestation, drought or tree pest infestation, fire, or a storm, among other causes. Remote sensing from satellites cannot readily distinguish between these drivers of land change. To correctly and consistently identify all drivers of land change (including the proximate causes and underlying drivers, [Lambin et al. 2001; Geist & Lambin 2002; Turner et al. 2007]), detailed understanding and on-the-ground observations of each individual change event would be required. This is extremely difficult as the occurrence of a land change only

becomes apparent after the change has happened, hence one cannot place observers at locations in advance. Identifying drivers of land change is a major undertaking, currently being attempted by several researchers (e.g. see [Lambin *et al.* 2003]. as an example for tropical regions), has been accomplished for a few specific drivers of land cover change and few land cover types (Zhu *et al.* 2016; Curtis *et al.* 2018), which we already highlighted in the discussion (line 220-221).

Overall, identifying drivers of land change globally and consistently is beyond the scope of our analyses presented in this manuscript. We have added a short definition of land change to the introduction (line 46-47).

In response to B): The reviewer suggests that biodiversity (e.g. species richness, abundance and community evenness) would be differently affected by temporary and permanent changes and suggests using land-cover time series to assess this. Although potentially methodologically relatively simple to do, this is problematic for several reasons:

- i) Land is changing constantly, through a combination of land cover and/or land use, therefore it is somewhat difficult to decide on what constitutes a permanent or a temporary land change. Are land changes lasting 12 months temporary while those lasting over 12 months permanent? An arbitrary cut off is problematic, as we analysed biodiversity data from a taxonomically broad range of species, ranging from soil invertebrates and annual plants to large mammals (Fig. 4). Therefore, a land change lasting several months may appear temporary to, for example, an elephant and will likely appear permanent to, for example, a mayfly (it's entire lifetime), given the different life spans of organisms.
- ii) Land cover maps capture change in land cover but not change in land use. Our metric of land change based on a spectral vegetation index captures changes in both, land cover and land use, recognizing them as a coupled system (Turner *et al.* 2007; Song *et al.* 2018). Therefore, using time series of land cover as suggested would only partially capture the abrupt land changes we analysed.
- iii) Land cover classifications have often modest accuracy for many land-cover classes. For example, the longest available global time series of land cover, the ESA Climate Change Initiative Land Cover dataset (Bontemps *et al.* 2013; ESA CCI 2017), has a classification accuracy of only 40% for grassland. Using the spectral data directly, instead of post-hoc classified data, as we have done in the analyses presented, avoids the classification inaccuracies of land cover products.
- iv) Time series of classified land cover are currently available for only 23 years while the Landsat time series that we used is available for 33 years (1992-2015 vs 1982-2015). Therefore, using land cover time series would limit our analyses, even when using the currently longest available land cover time series, namely the ESA Climate Change Initiative Land Cover data (ESA CCI 2017).

Furthermore, the reviewer's comment implies that differential impacts on communities are "expected" from land change caused by different drivers and temporary or permanent change. However, we are unaware of any global studies having documented such differences in biodiversity response and now highlight this as research gap in the discussion (line 220-221).

NB: from the examples mentioned by the reviewer, we would also like to note that both ‘permanent’ and ‘temporary’ changes can be caused by human (e.g., crop rotation, intensification, or reforestation) and/or natural (e.g., climatic anomalies, fire) drivers.

2) The authors replied to one of my comment on the uncertainty in coordinate precision:

"We assumed study authors are reporting site coordinates as precisely and accurately as possible."

I have experience with data collection of coordinates from field studies, and I believe this assumption cannot be made. 3000m is nothing when coordinates are expressed as decimal degrees with one decimal only, which is extremely common in field studies. Further, there are obvious rounding errors (rounding minutes in degree-minute-seconds is also quite common), note that field studies report coordinates for different purposes than being used in other analyses. This can be easily checked by plotting the coordinates provided by field studies for small islands or along the coast, which very often fall into the sea. To be on the safe side, I would suggest to repeat the analysis extending the buffer further than the MLE (it can be a sensitivity analysis).

3. We apologise for our misleading statement in our previous response. To clarify, we did not “assume[.] study authors are reporting site coordinates as precisely and accurately as possible” and we are fully aware of general issues around reported coordinate imprecision of field site locations.

Because of issues with coordinates reported by field studies, in this study we are using carefully curated site coordinates from the PREDICTS database. The PREDICTS database contains published studies for which the coordinates were obtained directly from the author(s). Given the importance of achieving the best possible coordinate accuracy, PREDICTS invested substantial time and effort in the curation of coordinates, employing a half-time GIS technician to check records, contact authors to get additional coordinates, and convert coordinates from study-specific coordinate systems to WGS84. For a description of the coordinate curation conducted by the PREDICTS project please see Hudson *et al.* (2014), specifically page 4709:

“We recorded each Site's coordinates as latitude and longitude (WGS84 datum), converting where necessary from local grid- based coordinate systems. Where precise coordinates for Sites were not available, we georeferenced them from maps or schemes available from the published sources or provided by authors. We converted each map to a semi- transparent image that was georeferenced using either ArcGIS (Environmental Systems Research Institute (ESRI) 2011) or Google Earth (http://www.google.co.uk/intl/en_uk/earth/), by positioning and resizing the image on the top of ArcGIS Online World Imagery or Google Maps until we achieved the best possible match of mapped geographical features with the base map. We then obtained geographic coordinates using geographic information systems (GIS) for each Site center or point location. We also recorded authors’ descriptions of the habitat at each Site and of any transects walked.”

And page 4710-1:

“Commonly encountered higher level problems included mistakes in coordinates, such as latitude and longitude swapped, decimal latitude and longitude incorrectly assembled from DD/MM/SS components, and direction (north/south, east/west) swapped round. These mistakes typically resulted in coordinates that plotted in countries not matching those given in the metadata and/or out to sea. The former was detected automatically by validation software, which required that the GIS-matched country for each Site (see “Biogeographical coverage” below) matched the country name entered in the PDF file for the Study; where a Study spanned several countries, we set the country name to “Multiple countries.” We visually inspected all Site locations on a map and compared them to maps presented in the source article or given to us by the authors, catching coordinates that were mistakenly out to sea and providing a check of accuracy.”

For 96.7% of all sites in the PREDICTS database precise coordinates were obtained from the publication or the author supplied data (based on an analysis of the “Coordinates_methods” column in the PREDICTS database). For the remaining 3.3% of PREDICTS studies coordinates were worked out by using information provided in the publications.

Furthermore, before conducting the analyses presented in our manuscript, we performed additional checks as described in the methods (line 278-282) and as a precaution “removed four studies with sites located in water (rivers, coastal areas or ponds), identified by intersecting all sites with a global permanent water surface mask (Pekel et al. 2016), as a precaution as sites within these studies likely have low positional accuracy”.

In summary, we have no evidence nor reasons to believe that there is any systematic bias – neither taxonomically nor geographically – in the coordinate accuracy or precision that could substantially affect our findings.

3) The response "in most situations the number of grid cells contributing to the mean EVI extracted from the Landsat imagery is identical for rectangular and circular buffers" and figure shown in relation to my comment on the rectangular buffer are misleading, as the effect is entirely dependent on the resolution used and the size of the buffer. The larger the buffer and the higher the resolution, the more problematic is the use of a rectangular buffer. That said, the author did not respond to my question. Why a rectangular buffer? A circular buffer is in principle a better approximation.

4. The reasons why we have used rectangular, or more accurately square, buffers are outline below. We now refer to these as “square buffers” (line 283) as this is a more accurate description as their sides are of equal length as given by the Maximum Linear Extent (MLE) for each study:

- i) square buffers likely capture the sampled sites better as the Maximum Linear Extent (MLE) was based on mostly rectangular design of sampling (see Figure in supplementary material to Hudson et al. 2014, reproduced here as Fig. R1). Examples of study sites are batches of line transects, vegetation quadrats, a set of pitfall traps or point counts distributed in a square area, where the coordinates provide the centroid of the rectangle. The sampling layouts of the majority of studies included in the PREDICTS database (see Fig. R1) suggest that a square buffer is likely more

accurately capturing the area sampled, contrary to the reviewer’s suggestion that “a circular buffer is in principle a better approximation”.

Fig. R1: Maximum linear extents of sampling. The thick black line indicates the distance that was recorded as Maximum Linear Extent (MLE). Reproduced from Hudson et al. (2014) supplementary material.

- ii) square buffers capture a larger area (21.5%) than circular buffers (assuming circular buffers would be constructed with a diameter of MLE), and therefore provide a more conservative estimate of the mean land change surrounding the sampled sites.

Ultimately, any buffer of any shape or size around the site coordinates will always be an approximation of the area actually sampled and—importantly—the area actually used by individuals observed by each sample.

We have conducted additional analyses to assess whether “[t]he larger the buffer and the higher the resolution, the more problematic is the use of a square buffer”. Since we are working with Landsat data throughout, our pixel resolution does not vary among sites or years. Thus, we compared the mean monthly EVI values in square and circular buffers at all sites with respect to their size. We found that monthly EVI values in square and circular buffers are highly correlated (Fig. R2b, average Pearson’s correlation coefficient: 0.987 ± 0.022 SD, $n = 21,338$ sites). Furthermore, contrary to the reviewer’s prediction, the difference in EVI between square and circular buffers was less in buffers of larger size than in buffers of smaller size (Fig. R2c) although the effect of buffer area (as measured by MLE) was not statistically significant (linear model: $F_{(1,581)} = 2.17$, $r^2 = 0.002$, $p = 0.14$), confirming that buffer area does not significantly affect EVI differences between square and circular buffers.

We have inserted a sentence in the methods section providing a reason why we used square buffers (line 285-287).

Fig. R2: Assessing the influence of square and circular buffer shapes on the extracted EVI estimates. (a) histogram of sampling extent as measured by Maximum Linear Extent (MLE) for 21,338 sites. Vertical line indicates the median MLE (70 m, $Q_1 = 30$ m, $Q_4 = 200$ m). (b) Mean monthly EVI values for square and circular buffers ($n = 2.6$ million site/months combinations). 1:1 line shown in red. (c) Mean study-specific ($N=582$) log-ratio of the mean monthly EVI of square relative to a circular buffer plotted against the size of buffer (as measured by MLE).

4) (Irrespective of what other referees believe) I still find the interpretation of Fig. 2 and 3 unnecessarily complex. It doesn't convey a clear message at glance, it's a lot of information to process (symbols, colours, categories, variables...). To me, a figure that takes too long to be interpreted and require the reader to check the legend multiple times is not a good way to present the results. In my opinion, the aim of figures is not to provide all possible informations (tables are a better option for that) but to simplify and convey a clear message. Eventually, this is a choice of the authors, but I suggest them to think carefully what to show and how to make it as clear as possible, normally, less is better. Note that you are aiming for a multidisciplinary journal with a very wide audience with different backgrounds.

5. We appreciate that we are displaying substantial amounts of data in our figures, which will require a reader to invest time to understand these. At the suggestion of the associate editor, we have gathered feedback from three colleagues, Dr Valerie Kapos (United Nations Environment Programme World Conservation Monitoring Centre, Cambridge UK and Department of Zoology, University of Cambridge, Cambridge UK), Dr Piero Visconti (Institute of Zoology, Zoological Society of London, UK and International Institute for Applied System Analyses, Laxenburg, Austria) and Professor Joseph Alcamo (Sussex Sustainability Research Programme, University of Sussex, UK), and have made a number of changes.

Based on colleagues' suggestions, we have simplified Figure 1, showing shifts in magnitude. A comparable map of shifts in trend is provided as Supplementary Fig. 1. We have added in-figure legends to Figures 1, 2 and 4 explaining the meaning of the colours and symbols to help readers' understanding. In addition, we have simplified some axis labels.

We have not amended Figure 3 as our three colleagues and ourselves think that the compositional similarity plots (Fig 3 a & b) and dendrograms (Fig 3 c & d) provide complementary information. This remains our preferred option for this figure, although we are willing to change this to alternatives (outlined below) if the associate editor deems these preferable. Alternatively, we could show only the dendrograms (Fig 3 c & d) in the main text, or show the compositional similarity plots (Fig 3 a & b) with matrices mirrored along the diagonal (see Fig R3 below, which may ease readability although duplicates information. Fig R3 is similar to what we showed previously (see Figure 2 in Newbold et al. 2015 *Nature* 520: 45-50). We would appreciate guidance from the associate editor about which of these visualization options would be preferable for Figure 3.

Fig. R3: Reduced compositional similarity between sites with and without an abrupt land change. Mean similarity in species assemblage composition (Sørensen similarity index) calculated between pairs of sites within the same study and land-use category without (UC) and with an abrupt land change of (a, c) varying shifts in magnitude, or (b, d) loss or gain in EVI (-/red and +/blue) and time passed between abrupt land change and biodiversity sampling (axis labels as in Fig. 2); Colours, from purple to brown (in a, b), indicate more or less similar assemblage composition with unchanged sites on average relative to comparisons among unchanged sites. Numbers (in a, b) indicate the total number of studies for which pairwise comparisons between sites could be made. All estimates are transformed relative to the compositional similarity between pairs of sites without a land change (UC – UC). (c,d) Dendrograms show hierarchical clustering of all pairwise similarities based on the average Manhattan distance between pairs of sites; sites with more similar assemblage composition are in branches of closer proximity.

Remember to change line 251

6. We have changed the information, now indicating the address at which we will make the code available (line 266-267).

365: add space

7. Good suggestion. We have added ‘spatial’ to this sentence, now stating ‘spatial pairs of sites’ (line 382).

Reviewer #3 (Remarks to the Author):

Thanks to the authors for clarifying and addressing all my comments and concerns. I think that the manuscript can be now accepted for publication as it is.

All the best,

Adrián.

8. We like to thank Adrián Regos for reviewing our revised manuscript, his helpful comments and positive feedback. We have added his name to the acknowledgements (line 570).

References

- Bontemps S et al. 2013. Consistent Global Land Cover Maps for Climate Modeling Communities: Current Achievements of the ESA’s Land Cover CCI. ESA Living Planet Symposium, Edimburgh **2013**:9–13. Available from https://ftp.space.dtu.dk/pub/Ioana/papers/s274_2bont.pdf.
- Curtis PG, Slay CM, Harris NL, Tyukavina A, Hansen MC. 2018. Classifying drivers of global forest loss. *Science* **361**:1108–1111. Available from <http://www.sciencemag.org/lookup/doi/10.1126/science.aau3445>.
- ESA CCI. 2017. ESA CCI Product User Guide Ver. 2. Available from <https://www.esa-landcover-cci.org/>.
- Geist HJ, Lambin EF. 2002. Proximate Causes and Underlying Driving Forces of Tropical Deforestation. *BioScience* **52**:143–150.
- Hudson LN et al. 2014. The PREDICTS database: a global database of how local terrestrial biodiversity responds to human impacts. *Ecology and Evolution* **4**:4701–4735. Available from <http://doi.wiley.com/10.1002/ece3.1303> (accessed December 4, 2014).
- Hudson LN et al. 2017. The database of the PREDICTS (Projecting Responses of Ecological Diversity In Changing Terrestrial Systems) project. *Ecology and Evolution* **7**:145–188. Available from <http://doi.wiley.com/10.1002/ece3.2579>.
- Lambin EF et al. 2001. The causes of land-use and land-cover change: moving beyond the myths. *Global Environmental Change* **11**:261–269. Available from <https://linkinghub.elsevier.com/retrieve/pii/S0959378001000073>.
- Lambin EF, Geist HJ, Lepers E. 2003. Dynamics of Land-use and Land -Cover Change in Tropical Regions. *Annual Review of Environment and Resources* **28**:205–241. Available from

- <http://www.annualreviews.org/doi/abs/10.1146/annurev.energy.28.050302.105459>.
- Newbold T et al. 2015. Global effects of land use on local terrestrial biodiversity. *Nature* **520**:45–50. Available from <http://www.nature.com/doi/abs/10.1038/nature14324>.
- Pekel J-F, Cottam A, Gorelick N, Belward AS. 2016. High-resolution mapping of global surface water and its long-term changes. *Nature* **540**:418–422. Nature Publishing Group. Available from <http://www.nature.com/doi/abs/10.1038/nature20584>.
- Song X-P, Hansen MC, Stehman S V., Potapov P V., Tyukavina A, Vermote EF, Townshend JR. 2018. Global land change from 1982 to 2016. *Nature* **560**:639–643. Springer US. Available from <http://www.nature.com/articles/s41586-018-0411-9>.
- Turner BL, Lambin EF, Reenberg A. 2007. The emergence of land change science for global environmental change and sustainability. *Proceedings of the National Academy of Sciences* **104**:20666–20671. Available from <http://www.pubmedcentral.nih.gov/articlerender.fcgi?artid=3001449&tool=pmcentrez&rendertype=abstract>.
- Zhu Z et al. 2016. Greening of the Earth and its drivers. *Nature Climate Change* **6**:791–795. Available from <http://www.nature.com/doi/abs/10.1038/nclimate3004>.